



# The VLF transmitters' radio wave anomalies related to 2010 Ms 7.1 Yushu earthquake observed by DEMETER satellite and the possible mechanism

Shufan Zhao[1*], XuHui Shen[1], Zeren Zhima[1] and Chen Zhou[2]

[1] Institute of Crustal Dynamics, China Earthquake Administration, Beijing 100085, China

[2] School of Electronic Information, Wuhan University, Wuhan, 430072, China

* Correspondence: zsf2008bj@126.com

**Abstract:** Earthquakes may disturb the lower ionosphere through various coupling mechanisms during their seismogenic and coseismic periods. The VLF signal radiated from ground-based transmitters will get affected when it penetrates the disturbed region in the ionosphere above the epicenter area, and this anomaly can be recorded by low earth orbit satellite under certain conditions. In this paper, the temporal and spatial variation of the Signal to Noise Ratio (SNR) of the VLF transmitter signal in the ionosphere over the epicenter of 2010 Yushu Ms 7.1 earthquake in China is analyzed. The results show that the SNR over the epicenter of Yushu earthquake especially in the southwestern region decreased (or dropped)revealed by one satellite revisit period before the main shock, which is consistent with the observed TEC anomaly at same time, implying that the decrease of SNR might be caused by the enhancement of TEC. A full-wave method was used to study the mechanism of the change of SNR before the earthquake. When the electron density in the lower ionosphere increases by four times, the electric field will decrease about 1 dB, indicating that the disturbed electric field decrease 20% compared with the original electric field and vice versa. It can be concluded that the variation of electron density before earthquakes may be one important factor influence the variation of SNR.

**Keywords:** 2010 Yushu earthquake, DEMETER satellite, VLF radio wave, signal to noise ratio, lower ionospheric disturbance, Full-wave model

## 1. Introduction

The VLF (Very Low Frequency) radio waves radiated by the powerful ground-based VLF transmitters have been utilizing for long distance communication and submarine navigation, because of the efficient reflection within the earth-ionosphere waveguide. However, there is still a small fraction of the wave energy can leak into the higher ionosphere and magnetosphere after being absorbed by the lower ionosphere. The signals from transmitters observed by the LEO (Low Earth Orbit) satellite can be used to research the propagation of VLF wave in the earth-ionosphere waveguide and ionosphere, as well as wave-particle interaction in the radiation belt (Cohen and Marshall, 2012; Inan et al., 2007; Inan and Helliwell, 1982; Lehtinen and Inan, 2009; Parrot et al., 2007).

It is gradually confirmed that earthquake abnormal precursors not only appear near the ground, but also may couple with the atmosphere and ionosphere through some mechanisms, resulting in plasma disturbances in the ionosphere and recorded by such as ionosonde and GPS-TEC (Global positioning system-Total electron


content) (Liu et al., 2009; Liu et al., 2001; Liu et al., 2006; Pulinets et al., 2000; Stangl et al., 2011; Zhao et al.,
2008). Therefore, when the VLF radio waves propagate upward through the earthquake affecting ionospheric
disturbance area, the VLF signals received by the ground VLF receivers and satellite will change, which have
been recorded by (Hayakawa, 2007; Maurya et al., 2016; Molchanov et al., 2006; Píša et al., 2013). In these
research, Molchanov et al. (2006) have applied the VLF data recorded by satellite to study the earthquake
abnormality for the first time. They found the SNR (Signal to Noise Ratio) of the electric field from VLF
transmitters recorded by DEMETER (Detection of Electro-Magnetic Emission Transmitted from Earthquake
Regions) satellite decreased near the epicenters during a series of earthquakes. The larger the magnitude of the
earthquake; the larger the spatial zone of SNR reduction is. However, it is hard to distinguish the coseismic
anomaly and precursory anomaly from their results.

Two devastating earthquakes, the 2008 Ms 8.0 Wenchuan earthquake and the 2010 Ms 7.1 Yushu

earthquake, have occurred successively in southwestern China during the operation period (2004-2010) of
DEMETER satellite. Some research also focused on the SNR variation of VLF transmitters using DEMETER
satellite observation to extract the earthquake related anomalies before the two strong earthquakes (He et al.,
2009; Shen et al., 2017; Yao et al., 2013). The results all illustrated the decrease of SNR before the earthquakes.
Since the seismo-ionospheric disturbance zone does not uniformly appear above the epicenter, the location of
the SNR abnormity in relation to the epicenter should be furtherly studied. Which factor influences the SNR and
its possible mechanism, that is also needed to be comprehensively illustrated. Therefore, in this paper we
investigate the temporal and spatial SNR variation of the VLF transmitter signal in the ionosphere over the
epicenter of 2010 Ms 7.1 Yushu earthquake, as well as the background variation of SNR in the same period of
2007-2010 to be distinguished whether the SNR reduction is caused by earthquake events or just ionospheric
background changes. The mechanism of how the seismo-ionospheric disturbance affect the variation of SNR
has been discussed in this paper.

As the mechanism of the VLF radio wave variations in the altitude of LEO satellite (presented as SNR

variation) before the earthquakes, Hayakawa (2007) and Píša et al. (2013) suggest the VLF anomalies exist
because the lower ionosphere is lowered before earthquake. Molchanov et al. (2006) declared that the variation
of SNR in the ionosphere is attributed to the plasma disturbance of the lower ionosphere. Furthermore, it has
been found that the electron density variation only exists in the lower ionosphere according to the computer
ionosphere tomography (CIT) results based on GPS-TEC data before Nepal Ms 8.1 earthquake in 2015 (Kong
et al., 2018). The electric field penetrating model of Kuo et al. (2011) shown that the electron density variation
in lower ionosphere can be induced by changes of the current in the global electric circuit which is also the
main factor making the lower ionosphere lowered. On the other hand, Marshall et al. (2010) construct a 3D
finite difference time domain model to simulate the lightning electromagnetic pulse and its interaction with the
lower ionosphere. All the reports mentioned above demonstrate that the earthquake and lightning can disturb
the electron density in the lower ionosphere. Many other studies also have found the main loss of VLF wave
power mainly occurs in the lower ionosphere when VLF radio waves penetrate into ionosphere through
calculations based on absorption curve or full wave model (Cohen and Inan, 2012; Liao et al., 2017; Starks et al.,
2008; Tao et al., 2010; Zhao et al., 2017; Zhao et al., 2015). In sum, the electron density variation in the lower
ionosphere might be one main factor causing SNR anomaly. Based on these results, the full-wave calculation
model was utilized to study the influence of the electron density disturbance of the lower ionosphere on the
variation of VLF radio signals.



In this paper, a brief description of the DEMETER data and full-wave method used in this study are
presented in Section 2. The temporal and spatial variations of SNR over the epicenter have been investigated
before 2010 Yushu earthquake with four years (2007-2010) observation; the full-wave model is used to
simulate how the variation of electron density in the lower ionosphere affects the electric field SNR of VLF
transmitter at the altitude of DEMETER are presented in Section 3. The discussion and conclusions of this
research are presented in Section 4 and 5.
**2. Materials and Methods**
*2.1. Earthquake, VLF Transmitters, and DEMETER data*
At the local time 07:49:37.9 of April 14, 2010, a Ms 7.1 earthquake hit the Yushu city, Qinghai Province
with epicenter is located in 33.2° N, 96.6° E with a 14 km depth at the Northeastern Tibetan plateau. The nearest
VLF transmitter around the epicenter is located in the proximity of Novosibirsk (NOV, in short) which belongs
to the Russian Alpha navigation system which consists of three transmitters. The other two transmitters named
Krasnodar (KRA) and Khabarovsk (KHA) are far away from Yushu earthquake, so only the satellite data
radiated from NOV have been used to analyze in this paper. The location of the transmitters and the epicenter of
Yushu earthquake are marked by blue squares and black stars respectively in Figure 1. Three different
frequency VLF radio signals (11.9/12.6/14.9 kHz) are radiated from these three transmitters, with a 0.4 s
duration and a 3.6 s cycle.
The DEMETER satellite was launched on 29 June 2004 as a sun-synchronous orbit at the altitude of 710
km, then was changed to 660 km in December 2005 (Parrot et al., 2006), and the mission was ended in
December 2010. The scientific objective of the DEMETER is to detect and characterize the electromagnetic
signals associated with natural phenomena (such as earthquakes, volcanic eruptions, tsunamis) or
anthropogenic activities. It operated in the region from invariant latitude -65° to 65°, with descending and
ascending orbits crossing the equator at local time ~10:00 and ~22:00, respectively. DEMETER has a re-visit
orbit period of about 14-days, which means the satellite returns to the over the same orbit trajectory after 13
days. The payloads include several electromagnetic sensors with two working mode: burst and survey. At
ELF/VLF band, the intensive electromagnetic wave data over locations of particular interest were provided in
the burst mode, and in the survey mode, electric and magnetic power spectral density (PSD) data every 2 s were
provided with sampling frequency 40 kHz and spectral resolution 19.53 Hz.
According to the formula of Dobrovolsky et al. (1979),the preparation zone of the earthquake can reach
$\rho=10^{0.43M}$,    *M* represents the magnitude of the earthquake in the formula. Considering the limited extension
of the Ms 7.1 Yushu earthquake, the preparation zone $\rho$ can reach to 1020 km, we mainly focused on the
region with radius 600 km above the epicenter to minimize the influence of other unknown disturbing sources.
In this study, the night-time PSD data of electric field from the DEMETER's survey mode observations
within the region of epicenter ±10° (black square in Figure 1) were extracted, and the data within 600 km
circle (shown in Figure 3) over the epicenter was used to study the perturbations of the VLF radio waves from
the Russia transmitters before and after the Yushu earthquake. The spectrum data of the 1st re-visit period
(April 2 to14) before the earthquake within 600 km are averaged shown as Figure 2 for example. It is clear
that the signals from many VLF transmitters can be distinguished. Due to the VLF radio signals at daytime is


too small to cause obvious SNR variation compared with that in night-time, we did not use the day-time data
in this study.

### 2.2. The method to calculate SNR

According to the method of Molchanov et al. (2006), the SNR of electric field was calculated as
follows:
$$SNR = \frac{2A(f_0)}{A(f_+) + A(f_-)} \quad (1)$$
where $A(f_0)$ is the amplitude of electric field spectrum at the central frequency, and $A(f_{\pm})$ are the
spectrums at $f_{\pm} = f_0 \pm \Delta f$, where $\Delta f$ is the chosen frequency band. As can be seen in Figure 2, the
VLF radio signals radiated from ground based VLF transmitters has different transmitting frequency
bands. For the three Russian VLF transmitters, the $f_0$ is set as three VLF radio waves frequency radiated
from NOV transmitters: 11.9/12.6/14.9 kHz, and the $\Delta f$=300 Hz.

### 2.3. Full wave method

A full-wave method has been used to seek a solution of Maxwell equations for plane waves varying as
$e^{jwt}$ in a horizontally-stratified medium with fixed dielectric permittivity tensors $\hat{\varepsilon}$ and permeability $\mu$ in
each layer. Considering the region of our interest is much smaller than the radius of the earth, we can neglect
the earth's curvature in the model of this study. A Cartesian coordinate system is established with x, y in the
horizontal plane and z vertical upward. The solution of the Maxwell equations is given in a form of a linear
combination of plane waves $\sim e^{j(k_\perp \cdot r_\perp)}$, where $k_\perp$ is the horizontal component of the wave vector k which
is conserved by Snell's law inside each layer, we have
$$\begin{cases} k \times E = \omega \mu_0 H \\ k \times H = -\omega \hat{\varepsilon} E \end{cases} \quad (2)$$
Where $\omega$ is the angular frequency, $\mu$ is the permeability of the medium ($\mu \equiv 1$ for non-magnetic
medium), $\hat{\varepsilon} = \varepsilon_0(I + \hat{\chi})$ is dielectric tensor, and $\hat{\chi}$ is electric susceptibility tensor (Yeh and Liu, 1972). $\hat{\chi}$ is
determined by the electron density and collision frequency in the ionosphere, as well as the geomagnetic field.
In our simulation, the electron density is obtained from International Reference Ionosphere (IRI) model, and
the electron collision frequency (denoted by $v$) is calculated by the exponential decay law with the height
(denoted by $h$) increasing $v=1.8\times10^{11}e^{-0.15h}$. The parameters of geomagnetic field at the location of the VLF
transmitter is calculated by International Geomagnetic Reference Field (IGRF) model.
Eliminating the z components from equation (2), the following elegant form of Maxwell equations are
obtained:



$$\frac{dV}{dz} = jk_0\hat{T} \cdot V \tag{3}$$
Where $V = (E_\perp, Z_0 H_\perp)$, $Z_0$ is wave impedance, $\hat{T}$ is a 4×4 matrix:
$$\hat{T} = \begin{pmatrix} -\frac{k_x\varepsilon_{31}}{k_0\varepsilon_{33}} & -\frac{k_x\varepsilon_{32}}{k_0\varepsilon_{33}} & \frac{k_xk_y}{k_0^2\varepsilon_{33}} & 1-\frac{k_x^2}{k_0^2\varepsilon_{33}} \\ -\frac{k_y\varepsilon_{31}}{k_0\varepsilon_{33}} & -\frac{k_y\varepsilon_{32}}{k_0\varepsilon_{33}} & -1+\frac{k_y^2}{k_0^2\varepsilon_{33}} & -\frac{k_xk_y}{k_0^2\varepsilon_{33}} \\ -\varepsilon_{21}+\frac{\varepsilon_{23}\varepsilon_{31}}{\varepsilon_{33}}-\frac{k_xk_y}{k_0^2} & -\varepsilon_{22}+\frac{\varepsilon_{23}\varepsilon_{32}}{\varepsilon_{33}}+\frac{k_x^2}{k_0^2} & -\frac{k_y\varepsilon_{23}}{k_0\varepsilon_{33}} & \frac{k_x\varepsilon_{23}}{k_0\varepsilon_{33}} \\ \varepsilon_{11}-\frac{\varepsilon_{13}\varepsilon_{31}}{\varepsilon_{33}}-\frac{k_y^2}{k_0^2} & \varepsilon_{12}-\frac{\varepsilon_{13}\varepsilon_{32}}{\varepsilon_{33}}+\frac{k_xk_y}{k_0^2} & \frac{k_y\varepsilon_{13}}{k_0\varepsilon_{33}} & -\frac{k_x\varepsilon_{13}}{k_0\varepsilon_{33}} \end{pmatrix} \tag{4}$$
The electromagnetic field in each layer can be obtained in the $k$ (wave vector) domain by solving equation (3)
recursively (Budden, 1985; Lehtinen and Inan, 2008). More details of full-wave method is described in
Lehtinen and Inan (2008).
**3. Results**
*3.1. VLF signal analysis from DEMETER satellite*
To calculate the SNR of electric field, Firstly, we collected the DEMETER's observations 5 re-visit
periods before and 1 re-visit period after the earthquake in 2010 to study the evolution of SNR above the
epicenter of Yushu earthquake. The data of the same period in 2007-2010 during non-earthquake time were
also extracted to build the background trend of SNR as a reference to confirm the anomaly occurred during
the earthquake. In the first period from April 2 to 14, there are two magnetic storms occurred on April 4-7 and
April 11-12 and, the Dst index are -84 nT, -69 nT and Kp index are 7 and 5.3, respectively. To exclude the
influence of geomagnetic storms, we only selected the data during Kp<3 and Dst > -30 nT. Secondly the
spatial range of SNR was determined according to the formula of Dobrovolsky et al. (1979). The SNR
distributions of three VLF radio signals are selected within a radius 600km above the region of epicenter, as
shown in Figure 3, where the black star represents the epicenter of Yushu earthquake.
It can be found that the 1st re-visit period (April 2-14) before the earthquake, the SNR of VLF waves at
three frequency (11.9, 12.6, 14.9 kHz) all decrease dramatically over the earthquake compared with other
revisit periods. Furthermore, to make sure whether the SNR abnormal is caused by ionospheric background
variation or not, we analyzed the SNR variation of the same period in 2007-2010 (background time). As
shown in Figure 3, there are some missing orbital data due to the magnetic storms from April 2 to 14 and the
SNR in some days like Mar 28 and February 20 are very small in all orbit, which was because the transmitter
is turned off on these days. We excluded these data in the examination of the change of SNR in Figure 4 and 5.
To avoid the impact on the result because of lacking data, the mean value of all the data in the black circle of
every period have been obtained to get the time sequence shown in Figure 4. The black dash line represents
the occurred date of the earthquake in Figure 4. The black and red lines represent the average values in 5
periods before the earthquake and 1 period after the earthquake inside the circle of radius 600 km above the
epicenter in 2010 and background time, respectively. The change trends of SNR in background time and 2010
are the same except in the 1st period before the earthquake. The average value of SNR decreases in 1st period



before the earthquake in 2010 which is different from it in background time especially in 11.9 kHz. It means
the decrease of SNR in 2010 might be caused by Yushu earthquake.
Furthermore, we focus on the daily changes of SNR in a period before the earthquake, To detect
abnormal signals of the SNR variations, a quartile-based process is performed (Liu et al., 2009). The median
(M) of every successive 11 days of the SNR of the whole orbits has been calculated to check the deviation
between the SNR of the 12th day and the computed median (M). The lower (first) quartile (denoted as LQ in
short) and the upper (third) quartile (UQ in short) have been calculated to provide the information about the
deviation. Based on the assumption of the normal distribution of the SNR with the mean (m) and standard
deviation ($\sigma$), the expected value of M and LQ or UQ are equals to m and 1.34$\sigma$ ((Liu et al., 2009) and
reference therein). We set the lower boundary (LB in short), LB = M+2(M-LQ) and the upper boundary (UB
in short), UB= M+2(UQ-M) to find the SNR anomalies with a stricter criterion. Thus, if an observed SNR on
the 12th day is greater or smaller than its previous 11-day-based UB or LB, a positive or negative abnormal of
SNR will be identified. Figure 5 shows the time series of SNR at 11.9, 12.6, 14.9 kHz respectively. The red,
gray, and two black curves denote the observed SNR and associated median and upper/lower bound (UB/LB),
respectively. Blue and green sign represent the upper and lower anomalous days identified by the computer
routine, respectively. The LB and UB are constructed by the 1–11 previous days' moving median (M), lower
quartile (LQ), and upper quartile (UQ). There are no circles on the red, gray, black lines in some days because
the data have been excluded as mentioned above. As we can see in the figure 5, besides the lower anomalies
appeared on April 13 (one day before Yushu earthquake, the occurred time of Yushu earthquake denoted by
vertical dashed line in Figure 5) at all transmitting frequency, there are another three anomalies occurred on
March 29, April 8,and April 10 respectively. Previous research (ndicate the earthquake precursory usually
occurred within one week before earthquake, so the lower anomaly occurred on March 29 at 12.6 and 14.9
kHz may have nothing to do with Yushu earthquake.
We speculate that the anomalies of SNR may be related to the anomalies of electron density. To confirm
our conjecture, we used GPS-TEC MAP data distributed by CODE (Center for Orbit Determination in Europe)
to check out whether the Total electron content (TEC) showing similar anomalies. The resolution of TEC data
from CODE is 5°×2.5°, We use 11 days' sliding mean value of every grid as background, then we can get a
spatial distribution of background. Background ± 2×stand deviation is set as threshold (Upper bound and
Lower bound) to determine whether there have anomalies, if intraday value exceed the threshold represents
there have anomalies. We have reviewed the TEC anomalies of every day from April 2 to April 14 (which
means the duration of sliding background is from March 22 to April 13). The TEC anomalies only occurred
April 13, especially the anomalies are the most intensive at UT 6:00 which means only the SNR anomaly at
April 13 is possible earthquake precursory, the other two anomalies at April 8 and 10 may be caused by other
factors. The top panel of Figure 6 shows the TEC at 6:00 am UT on April 13 and the sliding mean of
background (April 2-12), the bottom panel shows the abnormal region where the TEC value exceed threshold
(background ± 2×stand deviation). As we can see that the TEC had abnormal enhancement on April 13 at
southwestern region of epicenter. Similar to this, the SNR of orbit No. 030939-1 on April 13 also decreased in
the southwestern direction in Figure 3. This phenomenon maybe illustrates the decrease of SNR caused by
TEC enhancement. Furthermore, this TEC enhancement was probably caused by earthquake, because it shows





very intensive conjugate response. However, TEC anomalies caused by geomagnetic storm do not exhibit this
kind of phenomenon generally(Zhao et al., 2008).

*3.2. The possible mechanism of SNR variation revealed by full-wave simulation*

In section 3.1, we analyzed the spatial and temporal characteristics of SNR during the five-revisit period
before and one revisit period after the Yushu earthquake. It can be found that the SNR decreased significantly
before the earthquake over the epicenter area of Yushu earthquake, especially in the southwestern direction.
After excluding the influence of geomagnetic storms, we furtherly explored the possible mechanism of SNR
abnormal variation in this section. As mentioned in the section 1, the electron density in the lower ionosphere
can be disturbed through various mechanisms before earthquakes. The electron density before Nepal
earthquake was obtained from computer ionosphere tomography method by using GPS data (Kong et al.,
2018). The left column (Figure 7 of (Kong et al., 2018)) is the variation of electron density at the height of
150 km. It can be seen that after UT 6:20, the electron density in the southwest of the epicenter decreases
significantly compared with that before 6:20, while the electron density in the northwest and southeast of the
epicenter increases significantly compared with that before 6:20, and the range of variation reaches about 30%.
However the electron density hardly change at the height of 450 km (see Figure 7 in (Kong et al., 2018)).
Marshall et al. (2010) have shown that 60 horizontal discharge pulses of 7 V/m can cause 50% change of
electron density in lower ionosphere, and 60 horizontal discharge pulses of 10 V/m can even cause 400%
change of electron density. Based on these results, the full-wave model was used to simulate the changes of
the electric field at satellite altitude excited by ground-based VLF transmitter caused by the enhancement or
decrease of electron density in the lower ionosphere, so as to furtherly determine the change law of SNR.

The full wave method (FWM) (Lehtinen and Inan, 2009) was utilized to simulate the electric field
between altitudes of 0 - 120 km induced by NOV transmitter which is the closest transmitter to epicenter of
Yushu earthquake. A Cartesian coordinate system was established with x, y in the horizontal plane and z
vertical upward.

We set a Gaussian shape perturbation at 110 km with 20 km bandwidth in the ionosphere. The
magnitude of the perturbation was set as maximum 4 times both increase and decrease compared to the
original electron density of nighttime (the average electron density above NOV transmitter during
20100402-20100414 at LT 22:00 calculated from IRI-2016 model). The perturbation patterns are shown in
the Figure 7. The other parameters of geomagnetic field and ionosphere are obtained from existing models
which have been introduced in section 2.3.

The electric filed only from ground surface to 120km have been calculated by full wave model, Because
the electromagnetic wave at VLF band will propagate upward as whistler mode. The group velocities of the
upward radiated whistler-mode are almost parallel, and these waves form a narrow collimated beam which
does not have much lateral spread. The direction of group velocities is determined by refractive index surface.
The refractive index surface of the upgoing whistler mode at 120km is shown in Figure 8. A ducted
propagation is adopted at this L shell (Clilverd et al., 2008) and the VLF wave power is spread in accordance
with the divergence of geomagnetic field lines with a linear reduction because the mode conversion (Lehtinen
and Inan, 2009; Shao et al., 2012).





The simulated results of electric field at 120 km height with different electron density along the
magnetic meridian plane within 1000 km area around the transmitter NOV are shown in Figure 9a. It can be
seen that the wave mode interference in the wave-guide has been mapped into the ionosphere in the electric
field (Lehtinen and Inan, 2009), and the electric field increases when the electron density decreases, and vice
versa (Figure 9a). Furthermore, the maximum value of the electric field varying with height is collected to
study the influence of the electron disturbance.
In the nighttime, it can be seen that when the electron density increases by four times, the maximum
electric field decreases about 1 dB at 120 km (see figure 9b). The variation is also 1 dB at DEMETER's
altitude (660 km) because of the linear reductions (Lehtinen and Inan, 2009; Shao et al., 2012), which implies
that the disturbed electric field decrease 20% compared with the original electric field (Figure 9b). In a short
time interval as a few days before the earthquake, the background noise can be assumed stable, so the change
of electric field can reflect the change of SNR. It can be concluded when the electron density increases by
four times, the variation of SNR is 20%. The simulated results illustrate that the variation of electron density
in the lower ionosphere before earthquake is one main factor of causing the abnormal the variation of SNR.
The more precise SNR variation needs more observation and simulation in the future.
**4. Discussion**
*4.1. The possible mechanism on how the earthquake induces the disturbance in the lower ionosphere*
Which coupling mechanism is effective to induce electron density anomalies in the D/E layer by
earthquakes is still an open question. Molchanov et al. (2006) declared the lower ionospheric disturbance is
caused by acoustic gravity wave triggered by earthquakes. At present, the coupling mechanism of electric
field proposed by Pulinets (2009) is widely accepted because it has been demonstrated by a series models
(Kuo et al., 2011; Namgaladze et al., 2013; Zhou et al., 2017) and observations (Gousheva et al., 2006;
Gousheva et al., 2008; Li et al., 2017). As for 2008 Wenchuan Ms 8.0 earthquake in China, Li et al. (2017)
reported continuous observations about the anomalous electric field which lasted longer but weaker than the
electric field induced by lightning during one month before Wenchuan earthquake. He suggests that the
abnormal electric field might be caused by the seismogenic activity of Wenchuan earthquake. Xu et al. (2011)
also found about 2 mV/m anomalous electric field in the F2 layer of ionosphere before the Wenchuan
earthquake. Gousheva et al. (Gousheva et al., 2006; Gousheva et al., 2008) revealed a large number of
anomalous electric fields before earthquakes using the Intercosmos satellite. In additional, it is demonstrated
that the anomalous electric field induced by earthquake could change the electron density in the lower
ionosphere by Kuo et al. (2011) and Zhou et al. (2017). Such as 2015 M 8.1Nepal earthquake, the electron
density variation was well explained by the ground electric field coupling model established by Zhou et al.
(2017).

*4.2. The other factors may induce disturbance in the lower ionosphere*
The lightning, geomagnetic storms and other natural sources may induce disturbance in the lower
ionosphere (Marshall et al., 2010; Maurya et al., 2016; Peter et al., 2006; Zigman et al., 2007). As known, the
intensive TEC change occurs during geomagnetic storms, and the change of TEC is affected intensively



during the main phase of the geomagnetic storm, gradually return to normal accompany with the recovery
phase. To avoid the effect of geomagnetic storms, the data which kp > 3 and Dst < -30 nT were excluded in
this research and the TEC anomaly detected in Figure 6 showed on one day after the recovery phase of
geomagnetic storm (top panel of Figure 10). Furthermore, the change pattern of TEC is totally different from
the one caused by earthquake, because the TEC anomalies caused by geomagnetic storm expand from
high-latitudes to mid-latitudes due to thermospheric neutral winds, E×B convection and so on (Pokhotelov et
al., 2008). From bottom panel of Figure 10, we can see the all the SNRs of whole orbit are large in April
5,6,7,11 during geomagnetic storm, especially at the higher latitude. However, SNR pattern in April 13 is
totally different. In sum, The TEC anomaly on April 13 should be unconcerned with geomagnetic storm. The
lightning flash is very rare in our research region (only 4 events from Feb 2010 to Apr 2010, which can be get
from the search result of website (https://lightning.nsstc.nasa.gov/nlisib/nlissearch.pl?coords=?579,18)), so
the effect of lightning could be ignored in this study.
**5. Conclusions**

In this paper, the SNR of electric field from the Russian VLF transmitter observed by DEMETER satellite

was analyzed before and after 2010 Ms 7.1 Yushu earthquake. The VLF signals from Russian VLF transmitters
can be clearly observed at frequency of 11.9, 12.6, 14.9 kHz over the epicenter from the electric field spectrum
data. To determine whether the SNR variation is related to Yushu earthquake, the data in quiet space weather
conditions (kp $\leq$3 and Dst $\geq$ -30 nT) have been selected during five satellite revisit periods before the earthquake
and one revisit period after the earthquake. The result shows that the SNR decreased during one revisit period
before Yushu earthquake in all case. Our analysis on SNR variation also shows that the SNR in April 13 is
smaller than that in other days over the epicenter, and the decrease of SNR is the most intensive at the
southwestern region when we divide the space over the epicenter of earthquake into four regions. These results
are consistent with the TEC anomalies in Figure 6. In addition, we also analyzed the SNR changes over the
epicenter in the same period from 2007-2010 as background map and found that the SNR changes trend of one
revisit period before the earthquake relative to background time were contrary to those in 2010. The change
trend of SNR decreased in 2010 but increased in background time in the 1$^{st}$ revisit period before the earthquake.
The change trend of SNR is the same in other revisit period both in 2010 and background time. In sum, it can be
concluded that the SNR over the epicenter of Yushu earthquake decreases abnormally in one satellite revisit
period before the earthquake, especially in the southwestern region of the earthquake, which is consistent with
the observed TEC anomaly before the earthquake. The decrease of SNR before the Yushu earthquake may be
due to the enhancement of electron density.

The electron density in the lower ionosphere may change abnormally before earthquake through some

coupling mechanisms. The full wave simulation result on NOV, which is the nearest transmitter next to Yushu
earthquake, indicates that the electric field at the altitude of satellite will change when we add a disturbance on
electron density in the lower ionosphere. That is to say that the SNR of electric field will also change when the
background noise is considered to be invariable a few days before the earthquake. The simulated results show
that the SNR of electric field will decrease with the increase of electron density in the lower ionosphere; the
SNR will increase with the decrease of electron density in the lower ionosphere. It can be concluded that the
variation of electron density before earthquakes may be one important factor influence the variation of SNR.



We will continually explore the law of SNR change and verify the mechanism we proposed with more

seismic events, by utilizing the newly launched LEO electromagnetic satellite (China Seismo-Electromagnetic
Satellite) (Shen et al., 2018; Zhao et al., 2019) in next work.


**Data Availability**
The DEMETER satellite data were provided by DEMETER scientific mission center
(http://demeter.cnrs-orleans.fr). The GPS-TEC data were provided by CODE (Center for Orbit Determination
in Europe) and can be downloaded from the website ftp://cddis.gsfc.nasa.gov/pub/gps/products/ionex.
**Author Contributions:** Conceptualization, S.Z.; Formal analysis, S.Z.; Investigation, S.Z.; Methodology, S.Z.,
R.Z., and X.S.; Resources, S.Z., X.S. and R.Z.; Supervision, S.Z.; Visualization, S.Z.; Writing-original draft,
S.Z.; Writing-review & editing, S.Z., R.Z, .C.Z., and X.S..
**Corresponding author**
Correspondence to Shufan Zhao.
**Competing interests**
The authors declare that they have no competing interests.
**Funding:** This work is supported by the National Science Foundation of China (Grant No. 41704156,
41574139, 41874174), National Key R&D Program of China (Grant No. 2018YFC1503501), the Special Fund
of the Institute of Earthquake Forecasting, China Earthquake Administration (Grant No:
2015IES010103,2018CSES0203)and the APSCO Earthquake Research Project Phase II.
**Acknowledgements**
We acknowledge the DEMETER scientific mission center for providing data of DEMETER satellite
(http://demeter.cnrs-orleans.fr). The GPS-TEC data were provided by CODE (Center for Orbit Determination
in Europe) and can be downloaded from the website ftp://cddis.gsfc.nasa.gov/pub/gps/products/ionex.

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



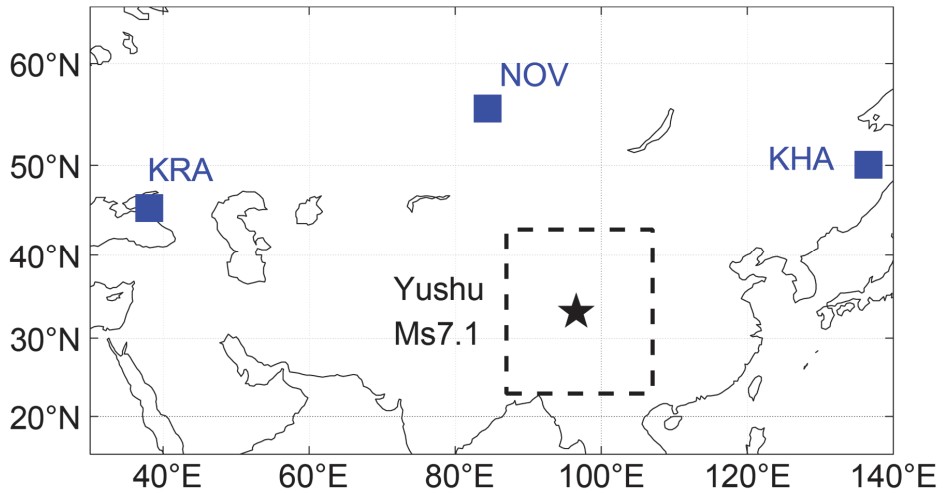

**Figure 1:** The locations of transmitters and Yushu earthquake. The blue squares represent the locations of the three transmitters (KRA, NOV, KHA) in Russia. The epicenter of Yushu earthquake is denoted by the black star. The black square covers the region of epicenter ±10° in which the data has been studied.



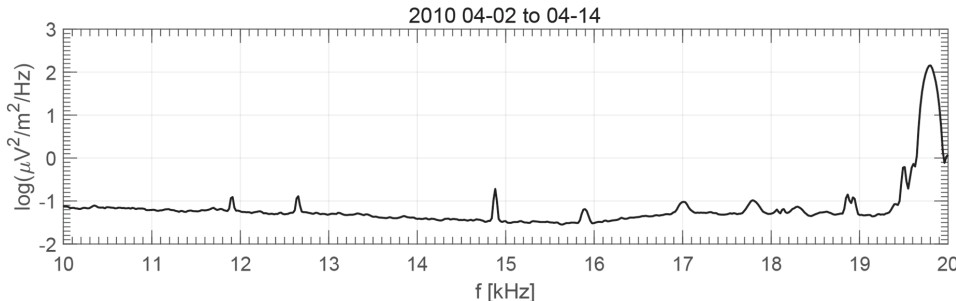


**Figure 2:** The averaged electric field spectrum structure within 600 km around the epicenter of Yushu earthquake.





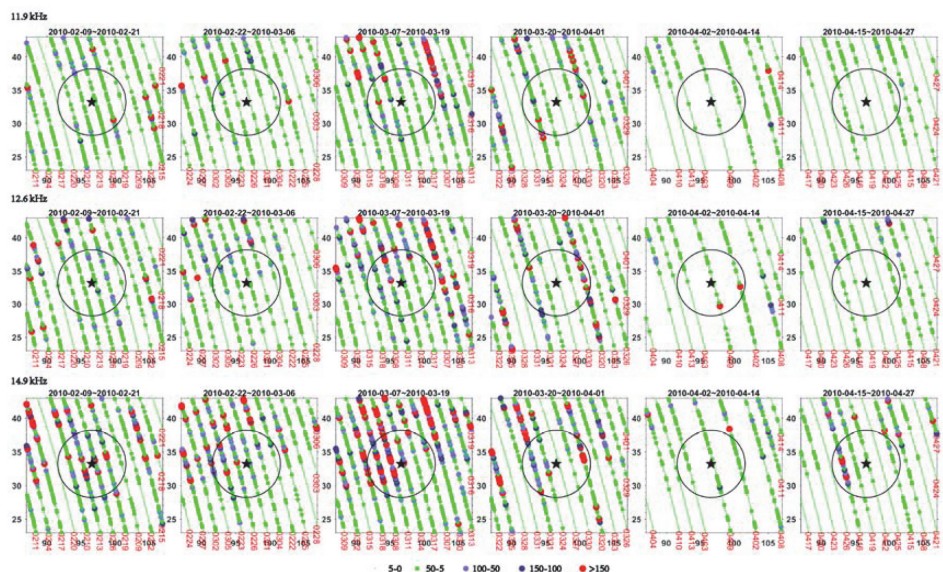


**Figure 3: The evolution of SNR evolution VLF radio waves frequencies 11.9, 12.6, 14.9 kHz with Δf = 300 Hz. The black star stands for the epicenter of the Yushu earthquake, and the radius of the black circle is 600km.**







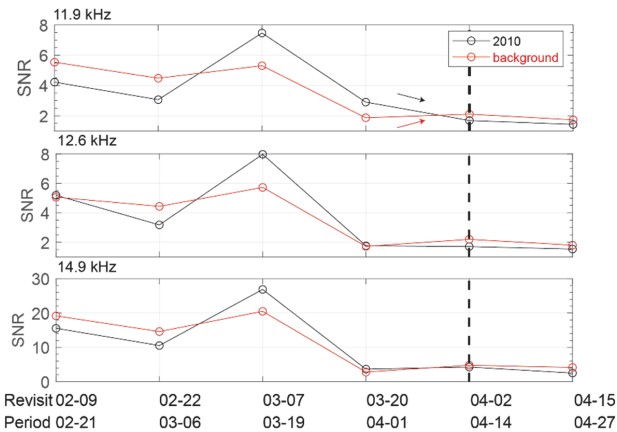


**Figure 4: The average SNR variation with revisit period inside the 600 km circle with the center of the epicenter. The panel from top to the bottom are the SNRs at 11.9, 12.6, 14.9 kHz and the numbers of the averaged data points. The green and red lines represent the SNR variations in 2010 and background time separately. The black dashed line represents the period with the end date of main shock date.**





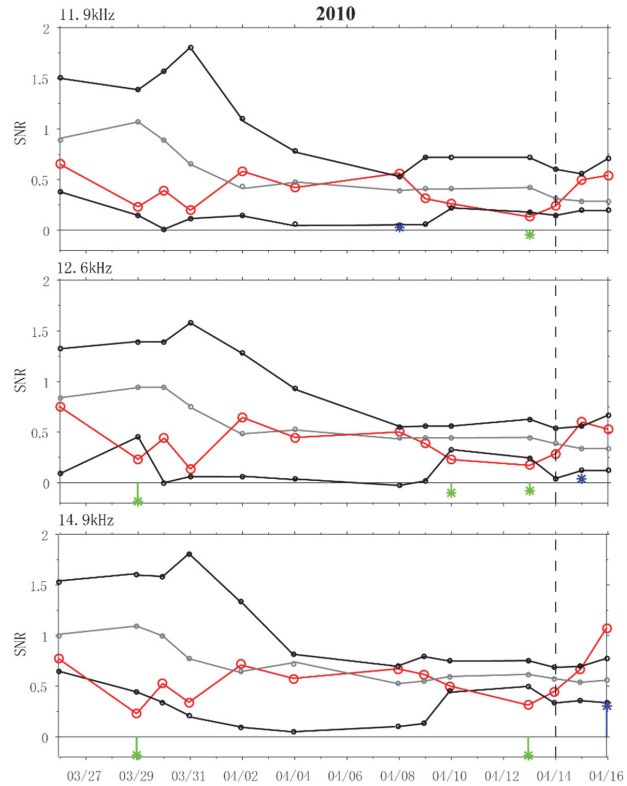


**Figure 5: A time series of SNR right above the Yushu epicenter. The Ms 7.1 Yushu earthquake occurred at**
**the local time 07:49:37.9 of April 14, 2010. The red, gray, and two black curves denote the observed SNR and**
**associated median and upper/lower bound (UB/LB), respectively. Blue and blue sgin represent the upper and**
**lower anomalous days identified by the computer routine, respectively. The LB and UB are constructed by**
**the 1–11 previous days' moving median (M), lower quartile (LQ), and upper quartile (UQ).**

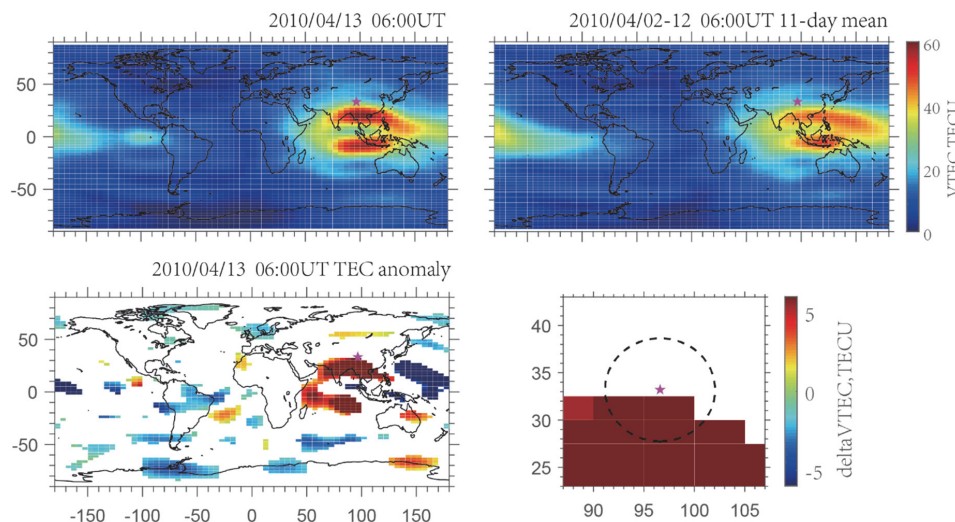

Figure 6 : The spatial distribution of GPS-TEC MAP (top) and its anomalies (bottom). The GPS-TEC MAP
on April 13 at UT 6:00 (left of top panel). The sliding mean of 11 days of background (right of top panel). The
global anomalies in GPS-TEC MAP (left of bottom panel). The regional anomalies around epicenter of
Yushu earthquake in GPS-TEC MAP (right of bottom panel). The purple pentagram indicate the epicenter
and the radius of the black circle is 600km.



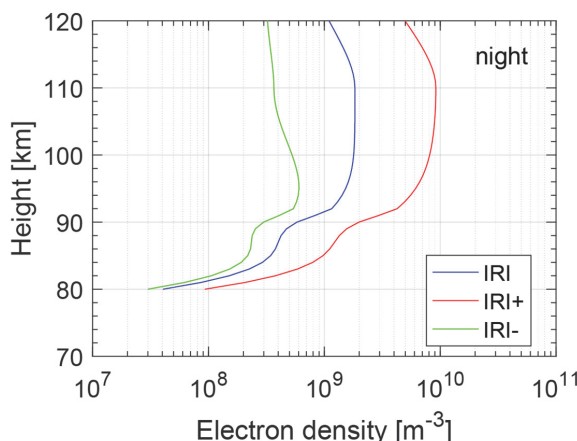

**Figure 7: The electron density profiles during night time. IRI represents the original electron density predicted by IRI model; IRI+ represents the electron density added Gaussian shape perturbation; IRI- represents the electron density subtracted Gaussian shape perturbation.**





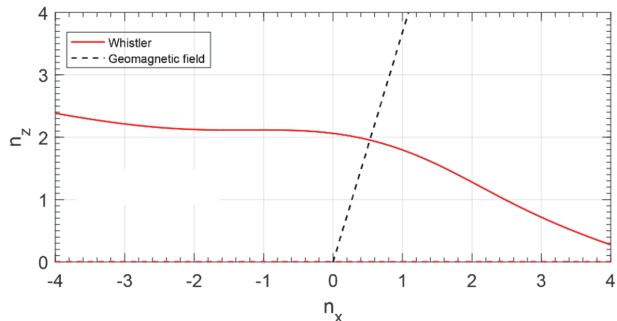

512

**Figure 8: The refractive index surface at 120 km. Red line shows a slice of the refractive index surface at**
**$n_y = 0$ of the whistler mode, calculated for $f = 11.9$ kHz at the altitude of $h = 120$ km. Black dash line**
**shows the direction of the geomagnetic field.**

516



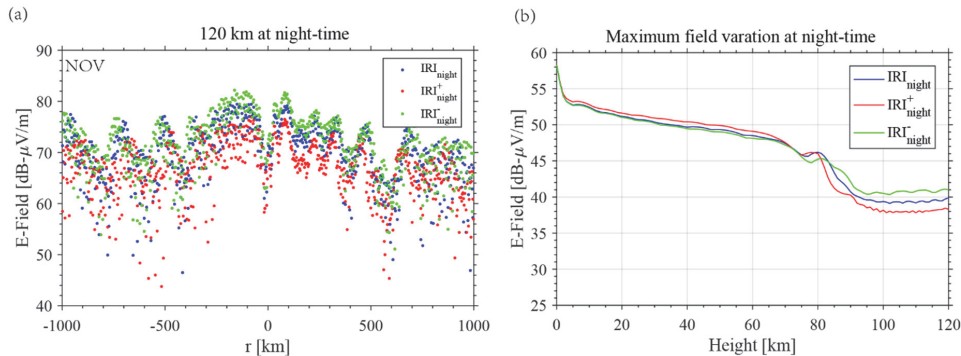

**Figure 9: The total electric field excited by ground based VLF transmitter NOV with transmitting frequency f=11.9 kHz and power P=500 kW. (a) The total electric field at the altitude of 120 km in the nighttime when Gaussian shape disturbance is set at 80-120km. (b) The maximum electric field varying with altitude in the nighttime.**



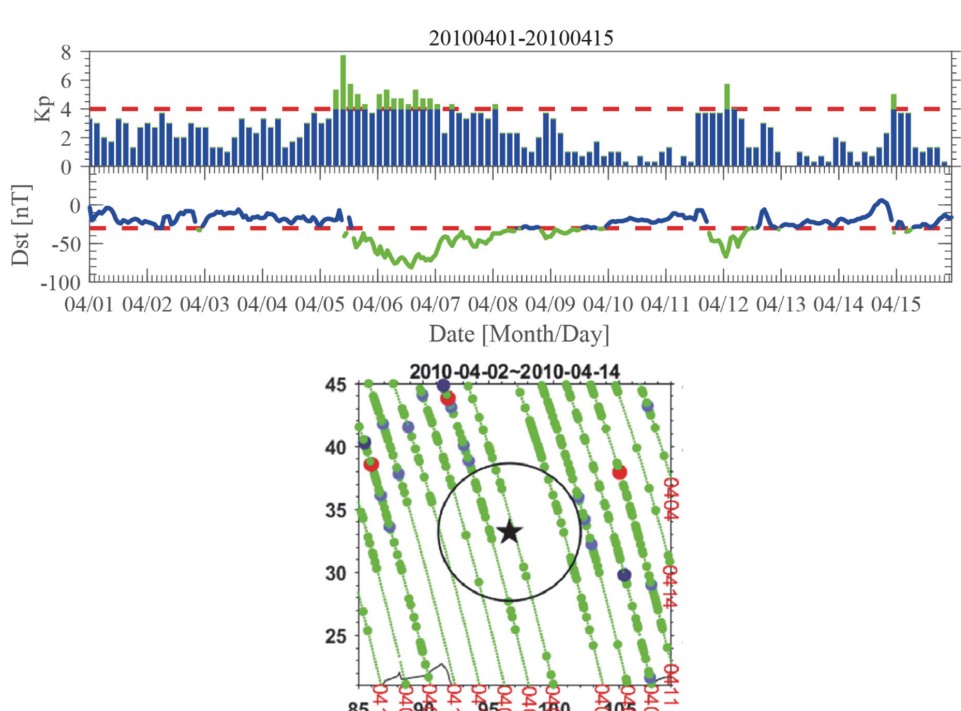

**Figure 10: The Kp and Dst index in April 2010 (top panel). The SNR distribution from April 2 to April 14.**
**(bottom panel)**