# Peer review of "The VLF transmitters' radio wave anomalies related to 2010"

_Annales Geophysicae, 2020_

## Referee Comment (RC1) · Anonymous Referee #1 · 6 Apr 2020

The article "The VLF transmitters' radio wave anomalies related to 2010 Ms 7.1 Yushu earthquake observed by DEMETER satellite and the possible mechanism" presents a detailed study of possible VLF precursor signals detected from satellite data. Various parameters are investigated: signal power at satellite location, vertical large scale TEC anomalies and a full wave model is used to investigate the effects in transmitted power by electron density variations in the E region. The authors point out many of the uncertainties in understanding this complex phenomenon and indicate the need of further studies, especially with the new CSES satellite. Some part of the text are difficult to understand, due to poor English sentences, therefore I have to indicate now major revisions, need to clarify and correct many important points.

The major scientific limitations, that should be discussed deeper, are, in my opinion:

- the day-to-day variability of the ionosphere, not separable from the average signal in the revisit cycle of DEMETER satellite.

- the lack of ground-based ionosonde data to support the hypotheses of the full wave simulations. These could also be from stations outside of the study area, that could provide an indication of the large-scale characteristics of the ionosphere.

- the effects of the geomagnetic storms occurring during April 2010 could also be studied on experimental data in the region nearby the event.

- Some explanation about how a positive TEC anomaly could be linked to a reduction in SRN, while the full wave model is limited to electron density profile in the E region. Most of the TEC seen by GPS is around the peak of F2 layer.

- Ducted VLF propagation paths could be studied in the region around the epicentre, to understand if the observed TEC anomaly on April 13, 2010 can have an impact of the VLF SNR.

I indicate in the following suggestions of corrections/improvements. This list is not exhaustive, additional careful check of the whole text in needed.

line 26: I suggest to change "utilising" into "used"

line 27 correct "the wave energy can" into "the wave energy that can"

line 28: I would use the word "absorbed" in the case when the signal is not propagated but absorbed through collisions between particles. In the case it is "refracted into"

line 31: Cohen and Marshall (2012) should not be cited in this sentence: the paper deals with ground observation, while this paragraph discusses VLF observed by LEO. It can be cited in this article, but in a different context.
line 35: Change "recorded by such as ionosonde and GPS-TEC" into "recorded by various instruments like ionosondes or GPS receivers measuring TEC"

line 41 and 52: avoid to use the word "abnormality" "abnormity", a better word can be "anomaly".

line 46: the Wenchuan earthquake is not studied in this paper. It could be explained that only the Yushu earthquake has been chosen for this study

line 99: correct "to the over the same" into "over the same".

line 105: the measurement unit for $\rho$ should be added.

line 109: figure 3 is referred before figure 2. I suggest to change the order of figures

line 112: add "and" between "averaged shown"

line 125: correct in the formula w into $\omega$

line 136-139: add references to IRI and IGRF versions used for this study and possibly also a reference to the electron collision frequency model.

lines145-147: the concept of numerical "swamping" could be explained in a few words, to illustrate the difficulties of full-wave modelling.

line 154: add the year to the calendar dates to avoid ambiguities, since in the previous sentence it was stated that data between 2007 and 2010 were considered.

line 155: add references and doing citations for Dst and Kp indices.

line 165: from the figures it seems that also during March 27 the VLF transmitter were not active.

line 193 correct the typo "(ndicate"

lines 220-224: I cannot access Kong et al, 2018 article, therefore I cannot see its Figure 7. I think it's better to avoid citing details of figures of another article, because readers

who can't access it, cannot follow the explanations. It also does not seems relevant to compare at that level the Nepal earthquake and Yushu earthquake.

lines 227-229: the work of Marshall et al. (2010) should be put in its context of simulation study and indicating the locations under consideration. The link between lightning activity and earthquake precursor electron density variations is not clear to me.

237: is 20 km a bandwidth or the sigma of the Gaussian curve?

244: correct "filed" into "field".

280: correct "In additional" into "In addition"

405: correct "gound" into "ground" and the page numbers in Marshal et al., 2010 reference.

Figure 3: since this figure is composed by many panels, their labels cannot be read without enlarging it on the screen. I suggest to use a bigger font size for the titles of each panels. The date of each track overlaps the longitude axis, making them difficult to read. This figure would benefit from plotting it full page in landscape mode, if this is possible on Annales Geophysicae. On this figure I do not understand if the range 0-5, which does not have a specified shape in the legend, indicates that there are no data, of if the SNR is so low that it is not clear if the signal is above the background noise level.

I suggest also to indicate in the caption that each row corresponds to a specific frequency and night-time observations. Add also that the date is indicated on the frame near the initial (or final?) point of the orbit pass. The passes when the VLF transmitters are not operating could also be indicated using a different graphical representation.

An additional comment out of curiosity: how the orbits during the geomagnetic storm are degraded with compared with the others? They could have been plotted on the figure, or on a supplementary material, by changing the graphical representation (e.g by plotting the orbital path in grey and fading the color of measured points).

Caption of figure 5, line 498: I suggest to add that the procedure to compute LB and UB is described in the text.
* * *

---

## Referee Comment (RC2) · Anonymous Referee #2 · 20 Apr 2020

Review of "The VLF transmitters◦Âű radio wave anomalies related to 2010 Ms 7.1 Yushu earthquake observed by DEMETER satellite and the possible mechanism" by Shufan Zhao et al.

The paper examines the potential effect of earthquake on VLF transmitter signal. Comparison of DEMETER observation and full wave simulation is made to demonstrate the hypothetic mechanism that earthquake induced enhancement of lower ionosphere density leads to the reduction of observed transmitter signal strength. Overall the results are interesting and potentially of great implication. While simulation results are reasonable, one challenging task in the manuscript is to establish that the SNR vari-

ation analyzed is related to Yushu earthquake. The reviewer has the following comments.

Regarding SNR calculation. The authors may want to examine the sensitivity of their results on the chosen delta_f to verify the variation of SNR is consistent. It may be useful to compare the variation of SNR within the circles marked in Figure 3 and that outside the circle to verify that the associated SNR variation is due to the earthquake.

line 39. incomplete sentence. maybe removing parentheses. line 40. Among those works, .... line 106. need unit for the preparation zone rho

---

## Author Comment (AC1) · 23 Apr 2020

First of all, we thank both of the reviewers for the careful reading and valuable comments on the manuscript. Our responses to the reviewer's comments are listed below one by one.

1. The major scientific limitations, that should be discussed deeper, are, in my opinion: the day-to-day variability of the ionosphere, not separable from the average signal in the revisit cycle of DEMETER satellite.

Reply: Thanks for the reviewer's suggestions. We will furtherly analyze the revisited

orbit data using moving average method to reduce the influence of the day-to-day variability of the ionosphere in the revised manuscript.

2. the lack of ground-based ionosonde data to support the hypotheses of the full wave simulations. These could also be from stations outside of the study area, that could provide an indication of the large-scale characteristics of the ionosphere.

Reply: Thanks for the reviewer's suggestions. Unfortunately, there is no ground-based ionosonde data acquired for us around the epicenter of Yushu earthquake. But we will check the COSMIC data to see it could support the full wave simulation or not. We will make this issue clear in the revised manuscript.

3. the effects of the geomagnetic storms occurring during April 2010 could also be studied on experimental data in the region nearby the event.

Reply: Thanks for the reviewer's suggestions. We have discussed the effects of geomagnetic storms on SNR in the section of discussion. And we will revise our manuscript to make it clearer.

4. Some explanation about how a positive TEC anomaly could be linked to a reduction in SRN, while the full wave model is limited to electron density profile in the E region. Most of the TEC seen by GPS is around the peak of F2 layer.

Reply: Thanks for the reviewer's suggestions. We plan to check the COSMIC data to find the anomaly of electron density in the E region and compare it with TEC anomaly. If there is no significant anomaly in E region, we will redo the full wave simulation by adding electron density perturbation in E/F region.

5. Ducted VLF propagation paths could be studied in the region around the epicentre, to understand if the observed TEC anomaly on April 13, 2010 can have an impact of the VLF SNR.

Reply: Thanks for the reviewer's suggestions. The results of Lehtinen et al. (Lehtinen et al., 2009) have shown that dominant peaks in the satellite data and in the calculated
field are not perfectly aligned which support that at lower altitudes (<1000 km) the propagation might be non-ducted; the same effect is seen in calculations of Starks et al. (2008) and Zhang et al. (2018). For the non-ducted, the direction of the group velocity (Vg) is not agree with the B0. But at higher altitude, the spreading of wave power is in accordance with the divergence of geomagnetic field line, where ducted propagation could be assumed. We will compare the abnormal region of TEC and SNR to study which model is dominated.

I indicate in the following suggestions of corrections/improvements. This list is not exhaustive, additional careful check of the whole text in needed.

Reply: Thanks for the reviewer's suggestions. We will check the whole text carefully.

line 26: I suggest to change "utilising" into "used" line 27 correct "the wave energy can" into "the wave energy that can"

Reply: Thanks for the reviewer's suggestions. I will revise these expressions in the revised manuscript.

line 28: I would use the word "absorbed" in the case when the signal is not propagated but absorbed through collisions between particles. In the case it is "refracted into"

Reply: Thanks for the reviewer's suggestions.

line 31: Cohen and Marshall (2012) should not be cited in this sentence: the paper deals with ground observation, while this paragraph discusses VLF observed by LEO. It can be cited in this article, but in a different context.

Reply: Thanks for the reviewer's suggestions. I am sorry about this mistake, which should be the error of my Endnote. Actually the article I cite is Cohen and Inan(2012) as you can see in the reference. I will check it carefully in the revised manuscript.

line 35: Change "recorded by such as ionosonde and GPS-TEC" into "recorded by various instruments like ionosondes or GPS receivers measuring TEC" line 41 and 52:

avoid to use the word "abnormality" "abnormity", a better word can be "anomaly".

Reply: Thanks. I will revise these expressions in the revised manuscript.

line 46: the Wenchuan earthquake is not studied in this paper. It could be explained that only the Yushu earthquake has been chosen for this study

Reply: Thanks for the reviewer's suggestions. I will revise the expressions.

line 99: correct "to the over the same" into "over the same". line 105: the measurement unit for _ should be added. line 109: figure 3 is referred before figure 2. I suggest to change the order of figures line 112: add "and" between "averaged shown"

Reply: Thanks. I will revise them in the manuscript.

line 125: correct in the formula w into !

Reply: Thanks. I will revise the formula.

line 136-139: add references to IRI and IGRF versions used for this study and possibly also a reference to the electron collision frequency model. lines145-147: the concept of numerical "swamping" could be explained in a few words, to illustrate the difficulties of full-wave modelling. line 154: add the year to the calendar dates to avoid ambiguities, since in the previous sentence it was stated that data between 2007 and 2010 were considered. line 155: add references and doing citations for Dst and Kp indices.

Reply: Thanks for the reviewer's suggestions. I will add more information in the manuscript.

line 165: from the figures it seems that also during March 27 the VLF transmitter were not active.

Reply: Thanks for the reviewer's suggestions. I will revise the expressions in the manuscript.

line 193 correct the typo "(ndicate"

Reply: Thanks. I will revise it in the manuscript.

lines 220-224: I cannot access Kong et al, 2018 article, therefore I cannot see its Figure 7. I think it's better to avoid citing details of figures of another article, because readers who can't access it, cannot follow the explanations. It also does not seems relevant to compare at that level the Nepal earthquake and Yushu earthquake.

Reply: Thanks for the reviewer's suggestions. I will revise the citation in the manuscript and try to use COSMIC data to find the anomalies of electron density in the D/E region before Yushu earthquake.

lines 227-229: the work of Marshall et al. (2010) should be put in its context of simulation study and indicating the locations under consideration. The link between lightning activity and earthquake precursor electron density variations is not clear to me.

Reply: Thanks for the reviewer's suggestions. The variation of electron density in the ionosphere caused by lightning activity and earthquake can both be explained and by one kind of Lithosphere-Atmosphere-Ionosphere Coupling mechanism, penetration of DC electric field (Zhou et al., 2017; Kuo et al., 2011). The results of Marshall et al. (2010) give the amplitude of the perturbation of the electron density in the D/E region caused by lightning flashes which provide us a reference on studying the earthquake.

237: is 20 km a bandwidth or the sigma of the Gaussian curve?

Reply: Thanks. It is the sigma of the Gaussian curve. I will revise this expression in the manuscript.

244: correct "filed" into "field". 280: correct "In additional" into "In addition" 405: correct "gound" into "ground" and the page numbers in Marshal et al., 2010 reference.

Reply: Thanks. I am sorry for these mistakes. I will revise in the manuscript.

Figure 3: since this figure is composed by many panels, their labels cannot be read without enlarging it on the screen. I suggest to use a bigger font size for the titles of

each panels. The date of each track overlaps the longitude axis, making them difficult to read. This figure would benefit from plotting it full page in landscape mode, if this is possible on Annales Geophysicae. On this figure I do not understand if the range 0-5, which does not have a specified shape in the legend, indicates that there are no data, of if the SNR is so low that it is not clear if the signal is above the background noise level. I suggest also to indicate in the caption that each row corresponds to a specific frequency and night-time observations. Add also that the date is indicated on the frame near the initial (or final?) point of the orbit pass. The passes when the VLF transmitters are not operating could also be indicated using a different graphical representation.

Reply: Thanks for the reviewer's suggestions. The legend of range 0-5 is so small which cannot be seen clearly, but it can be seen in the figure. I will modify this figure to make it clear for readers.

An additional comment out of curiosity: how the orbits during the geomagnetic storm are degraded with compared with the others? They could have been plotted on the figure, or on a supplementary material, by changing the graphical representation (e.g by plotting the orbital path in grey and fading the color of measured points).

Reply: Thanks for the reviewer's suggestions. I will modify the figure to show this phenomenon.

Caption of figure 5, line 498: I suggest to add that the procedure to compute LB and UB is described in the text.

Reply: Thanks for the reviewer's suggestions. I will add the description in the revised manuscript.

Reference

1. Lehtinen, N. G., and Inan, U. S.: Full wave modeling of transionospheric propagation of VLF waves. Geophys. Res. Lett. 36, 2009.

2. Starks, M., Quinn, R., Ginet, G., Albert, J., Sales, G., Reinisch, B., and Song, P.:

[Figure]

Illumination of the plasmasphere by terrestrial very low frequency transmitters: Model validation. J. Geophys. Res.: Space Phys 113, 2008.

3. Zhang, Z. X., Chen, L. J., Li, X. Q., Xia, Z. Y., Heelis, R. A., and Horne, R. B.: Observed Propagation Route of VLF Transmitter Signals in the Magnetosphere, J Geophys Res-Space, 123, 5528-5537, 2018.

4. Zhou, C., Liu, Y., Zhao, S. F., Liu, J., Zhang, X. M., Huang, J. P., Shen, X. H., Ni, B. B., and Zhao, Z. Y.: An electric field penetration model for seismo-ionospheric research. Advances In Space Research 60, 2217-2232, 2017.

5. Kuo, C. L., Huba, J. D., Joyce, G., and Lee, L. C.: Ionosphere plasma bubbles and density variations induced by pre-earthquake rock currents and associated surface charges. Journal of Geophysical Research Atmospheres 116, 2011

6. Marshall, R. A., Inan, U. S., and Glukhov, V. S.: Elves and associated electron density changes due to cloud-to-ground and in-cloud lightning discharges. Journal of Geophysical Research: Space Physics 115, 2010.

Please also note the supplement to this comment:
https://www.ann-geophys-discuss.net/angeo-2020-7/angeo-2020-7-AC1-supplement.pdf

---

## Author Comment (AC2) · 23 Apr 2020

First of all, we thank both reviewers for careful reading and valuable comments on the manuscript, which improve the quality of our manuscript. Our responses to the reviewer's comments are listed below one by one.

Regarding SNR calculation. The authors may want to examine the sensitivity of their results on the chosen delta_f to verify the variation of SNR is consistent. It may be useful to compare the variation of SNR within the circles marked in Figure 3 and that outside the circle to verify that the associated SNR variation is due to the earthquake.

[Figure]

Reply: Thanks for the reviewer's suggestions. We will test this and add the results in the revised manuscript.

line 39. incomplete sentence. maybe removing parentheses. line 40. Among those works, .... line 106. need unit for the preparation zone rho

Reply: Thanks for the reviewer's suggestions. We will revise these and check the whole text of manuscript carefully.

Please also note the supplement to this comment:
https://www.ann-geophys-discuss.net/angeo-2020-7/angeo-2020-7-AC2-supplement.pdf

---

## Author Response (AR1)

Cover Letter

Dear Editor Mirko Piersanti,

Enclose is the revised version of the paper entitle "The VLF transmitters' radio wave anomalies related to 2010 Ms 7.1 Yushu earthquake observed by DEMETER satellite and the possible mechanism" by Zhao et al.

We would like to thank you for the editorial consideration and for giving the opportunity to improve our manuscript. According to the reviewers' comments, substantial revision has been made in this revised version of the manuscript, and the reviewers' concerns have been addressed carefully point-by-point, as shown below.

The authors also would like to express the heartfelt gratitude to the reviewers for their insightful comments and valuable suggestions. The summarized results of main interest, general comments, questions and detailed comments benefit quite a lot our current revision, which aims to put the paper into a more suitable shape.

Based upon all the above, we resubmit the revised manuscript to the Journal for further evaluations and potential publication. All co-authors have agreed to the final version of this paper, as submitted herewith. If available, we would like to request the assignment of the same referees for reviews.

Thank you again for your editorial consideration.

Sincerely,

Shufan Zhao

Institute of Crustal Dynamics, China Earthquake Administration

Beijing , China

First of all, we thank both of the reviewers for the careful reading and valuable comments on the manuscript. Our responses to the reviewer's comments are listed below one by one.

Reviewer #1

1. The major scientific limitations, that should be discussed deeper, are, in my opinion: the day-to-day variability of the ionosphere, not separable from the average signal in the revisit cycle of DEMETER satellite.

Reply:

Thanks for the reviewer's suggestions. Furtherly, the daily variation of SNR in the 1st period before the earthquake is studied using a quartile-based process to detect the anomaly of the SNR. The result demonstrated there is a negative anomaly on April 13 at all transmitting frequency (shown in Figure 4). However, the result in Figure 4 which includes successive 20-day orbital data may be carried into the ionospheric background noise of different place. To avoid this kind of ionospheric background noise, we have furtherly analyzed the revisited orbit data using moving quartile-based process to reduce the influence of different place in the ionosphere. The results are shown in Figure 5, which also indicate the negative anomalies in April 13.

2. the lack of ground-based ionosonde data to support the hypotheses of the full wave simulations. These could also be from stations outside of the study area, that could provide an indication of the large-scale characteristics of the ionosphere.

Reply:

Thanks for the reviewer's suggestions. Unfortunately, there is no ground-based ionosonde data acquired for us around the epicenter of Yushu earthquake. But we use COSMIC data to see whether there is disturbance in the D / E layer of the ionosphere. The results are shown in Figure 7 which could support the hypotheses of the full wave simulations.

3. the effects of the geomagnetic storms occurring during April 2010 could also be studied on experimental data in the region nearby the event.

Reply:

Thanks for the reviewer's suggestions. We have discussed the effects of geomagnetic storms on SNR in the section of discussion in line 308-313.

4. Some explanation about how a positive TEC anomaly could be linked to a reduction in SRN, while the full wave model is limited to electron density profile in the E region. Most of the TEC seen by GPS is around the peak of F2 layer.

Reply:

Thanks for the reviewer's suggestions. We add some explanation in line 242-246.

5. Ducted VLF propagation paths could be studied in the region around the epicentre, to understand if the observed TEC anomaly on April 13, 2010 can have an impact of the VLF SNR.

Reply:
Thanks for the reviewer's suggestions. The results of Lehtinen et al. (Lehtinen et al., 2009) have shown that dominant peaks in the satellite data and in the calculated field are not perfectly aligned which support that at lower altitudes (<1000 km) the propagation might be non-ducted; the same effect is seen in calculations of Starks et al. (2008) and Zhang et al. (2018). For the non-ducted, the direction of the group velocity (Vg) is not agree with the $B_0$. But at higher altitude, the spreading of wave power is in accordance with the divergence of geomagnetic field line, where ducted propagation could be assumed. However, Cliverd et al.(2008) declared a ducted propagation is adopted at the L shell of Yushu earthquake. In this paper, we can see the abnormal region of TEC and SNR both located in the southwestern region of Yushu epicenter, which could demonstrated the VLF radio wave propagate in ducted mode.

I indicate in the following suggestions of corrections/improvements. This list is not exhaustive, additional careful check of the whole text in needed.

Reply:
Thanks for the reviewer's suggestions. We have checked the whole text carefully.

line 26: I suggest to change "utilising" into "used"
line 27 correct "the wave energy can" into "the wave energy that can"

Reply:
Thanks for the reviewer's suggestions. I have revised these expressions in the revised manuscript.

line 28: I would use the word "absorbed" in the case when the signal is not propagated but absorbed through collisions between particles. In the case it is "refracted into"

Reply:
Thanks for the reviewer's suggestions.

line 31: Cohen and Marshall (2012) should not be cited in this sentence: the paper deals with ground observation, while this paragraph discusses VLF observed by LEO. It can be cited in this article, but in a different context.

Reply:
Thanks for the reviewer's suggestions. I am sorry about this mistake. I have revised the manuscript.

line 35: Change "recorded by such as ionosonde and GPS-TEC" into "recorded by various instruments like ionosondes or GPS receivers measuring TEC"
line 41 and 52: avoid to use the word "abnormality" "abnormity", a better word can be "anomaly".

Reply:
Thanks. I have revised these expressions in the revised manuscript.

line 46: the Wenchuan earthquake is not studied in this paper. It could be explained that only
the Yushu earthquake has been chosen for this study
Reply:
Thanks for the reviewer's suggestions. I have revised in line 53-55.
line 99: correct "to the over the same" into "over the same".
line 105: the measurement unit for _ should be added.
line 109: figure 3 is referred before figure 2. I suggest to change the order of figures
line 112: add "and" between "averaged shown"
Reply:
Thanks. I have revised them in the manuscript.
line 125: correct in the formula w into !
Reply:
Thanks. I have revised the formula.
line 136-139: add references to IRI and IGRF versions used for this study and possibly also
a reference to the electron collision frequency model.
lines145-147: the concept of numerical "swamping" could be explained in a few words, to
illustrate the difficulties of full-wave modelling.
line 154: add the year to the calendar dates to avoid ambiguities, since in the previous
sentence it was stated that data between 2007 and 2010 were considered.
line 155: add references and doing citations for Dst and Kp indices.
Reply:
Thanks for the reviewer's suggestions. I have added these information in the manuscript.
line 165: from the figures it seems that also during March 27 the VLF transmitter were not
active.
Reply:
Thanks for the reviewer's suggestions. The explanation is given in line 150-156. In Figure 2
we depict the orbit during geomagnetic storm by hollow dots and blue fonts and depict the
orbit when the VLF transmitter turn off by grey color. But these orbital data in these days
have been excluded in the following analyses and figures.
line 193 correct the typo "(ndicate"
Reply:
Thanks. I have revised it in the manuscript.
lines 220-224: I cannot access Kong et al, 2018 article, therefore I cannot see its Figure 7. I
think it's better to avoid citing details of figures of another article, because readers who can't access it, cannot follow the explanations. It also does not seems relevant to compare at that
level the Nepal earthquake and Yushu earthquake.
Reply:
Thanks for the reviewer's suggestions. I have revised these contents in Line 231-233 and use
COSMIC data to find the anomalies of electron density in the D/E region before Yushu
earthquake. Yushu and Nepal earthquake both belong to intraplate earthquakes, caused by
collision between the Indian plate and Qinghai-Tibetan plate, and the magnitude is also
comparable. So, it could be possible to compare the ionospheric disturbance induced by the
Nepal and Yushu earthquake.
lines 227-229: the work of Marshall et al. (2010) should be put in its context of simulation
study and indicating the locations under consideration. The link between lightning activity
and earthquake precursor electron density variations is not clear to me.
Reply:
Thanks for the reviewer's suggestions. The variation of electron density in the ionosphere
caused by lightning activity and earthquake can both be explained and by one kind of
Lithosphere-Atmosphere-Ionosphere Coupling mechanism, penetration of DC electric field
(Zhou et al., 2017; Kuo et al., 2011). The results of Marshall et al. (2010) give the amplitude
of the perturbation of the electron density in the D/E region caused by lightning flashes which
provide us a reference on studying the earthquake. We have revised the text in Line 235-238.
237: is 20 km a bandwidth or the sigma of the Gaussian curve?
Reply:
Thanks. It is the sigma of the Gaussian curve. I have revised this expression in Line 251 of
the manuscript.
244: correct "filed" into "field".
280: correct "In additional" into "In addition"
405: correct "gound" into "ground" and the page numbers in Marshal et al., 2010 reference.
Reply:
Thanks. I am sorry for these mistakes. I have revised in the manuscript.
Figure 3: since this figure is composed by many panels, their labels cannot be read without
enlarging it on the screen. I suggest to use a bigger font size for the titles of each panels. The
date of each track overlaps the longitude axis, making them difficult to read. This figure
would benefit from plotting it full page in landscape mode, if this is possible on Annales
Geophysicae. On this figure I do not understand if the range 0-5, which does not have a
specified shape in the legend, indicates that there are no data, of if the SNR is so low that it
is not clear if the signal is above the background noise level. I suggest also to indicate in the
caption that each row corresponds to a specific frequency and night-time observations. Add
also that the date is indicated on the frame near the initial (or final?) point of the orbit pass.

The passes when the VLF transmitters are not operating could also be indicated using a different graphical representation.

Reply:

Thanks for the reviewer's suggestions. We have revised Figure 2 and its caption to make it clear for readers.

An additional comment out of curiosity: how the orbits during the geomagnetic storm are degraded with compared with the others? They could have been plotted on the figure, or on a supplementary material, by changing the graphical representation (e.g by plotting the orbital path in grey and fading the color of measured points).

Reply:

Thanks for the reviewer's suggestions. I have revised this figure.

Caption of figure 5, line 498: I suggest to add that the procedure to compute LB and UB is described in the text.

Reply:

Thanks for the reviewer's suggestions. I have added the description in the revised manuscript.

Reviewer #2

Regarding SNR calculation. The authors may want to examine the sensitivity of their results on the chosen delta_f to verify the variation of SNR is consistent. It may be useful to compare the variation of SNR within the circles marked in Figure 3 and that outside the circle to verify that the associated SNR variation is due to the earthquake.

Reply:

Thanks for the reviewer's suggestions. We exclude the circle and focus the data in the whole black square shown in Figure 1. Furtherly, we have tested some orbits which located at different distance away from epicenter. As we can see there is no obvious anomaly in the orbit of April 9 overhead the epicenter of Yushu earthquake and in the orbit of April 10 which is very far from the epicenter (as shown in Figure 5).

line 39. incomplete sentence. maybe removing parentheses. line 40. Among those works, .... line 106. need unit for the preparation zone rho

Reply:

Thanks for the reviewer's suggestions. We have revised these expressions and check the whole text of manuscript carefully.

[revised manuscript text omitted]